# Pastoralism may have delayed the end of the green Sahara

Chris Brierley [iD] [1], Katie Manning[2,3] & Mark Maslin[1]

The climate deterioration after the most recent African humid period (AHP) is a notable past example of desertification. Evidence points to a human population expansion in northern Africa prior to this, associated with the introduction of pastoralism. Here we consider the role, if any, of this population on the subsequent ecological collapse. Using a climate-vegetation model, we estimate the natural length of the most recent AHP. The model indicates that the system was most susceptible to collapse between 7 and 6 ka; at least 500 years before the observed collapse. This suggests that the inclusion of increasing elements of pastoralism was an effective adaptation to the regional environmental changes. Pastoralism also appears to have slowed the deterioration caused by orbitally-driven climate change. This supports the view that modern pastoralism is not only sustainable, but beneficial for the management of the world's dryland environments.

[1] Department of Geography, University College London, London WC1E 6BT, UK. [2] Institute of Archaeology, University College London, London WC1E 6BT, UK. [3] Department of Geography, Kings College London, London WC2R 2LS, UK. Correspondence and requests for materials should be addressed to C.B. (email: c.brierley@ucl.ac.uk)

Typically, traditional subsistence pastoralism has been seen as agents of environmental degradation through over-grazing, habitat change and resource competition with wildlife. This view (Fig. 1a) was embedded in the environmental doctrine of the twentieth century, partly as a consequence of the historical relationship between colonial administrators and traditional pastoralists[1]. This doctrine has led to a recent suggestion that early pastoralism was so unsustainable that it triggered a climatic deterioration in the Sahara around 5500 years ago[2] (at the end of the African Humid Period[3]). This has significant implications for the way in which modern populations living in marginal environments are perceived, and particularly how modern pastoralism is recognised within local and regional ecological and economic policies. This suggestion goes against research demonstrating the sustainability of pastoralism[4–6].

Tipping points and threshold behaviours are an emotive topic when talking about future climate change[7]. A common example is the African Humid Period (AHP) lasting from 14,700 years ago[8] to around 5500 years ago[3] (Fig. 2a), colloquially termed the "green Sahara". With the onset of favourable orbital conditions around 14.7 ka summer rains penetrated much further into northern Africa[8]. As a result, humid conditions were established initially at lower latitudes, and progressively later at more northern latitudes[3,9,10]. Pollen reconstructions[11] indicate a mix of tropical elements reaching up to 20° N, and Sudanian woodland and Sahelian grasslands extending at least as far as 28° N. These changes supported numerous Sahelian and aquatic animals, such

as elephant, crocodile and fish[12]. Yet, debate is on-going over the rate of climatic deterioration at the end of the humid period. Both sediment flux records from deep sea cores off the coast of north-west Africa[13–15] (Fig. 2a) and $\Delta D_{wax}$ isotopic values from east and northeast Africa[9,16] point to a rapid shift 5500 years ago. Pollen and sedimentological records from Lake Yoa in northern Chad, however, indicate a more gradual deterioration of the regional ecosystem[17,18] (Fig. 2a). This discrepancy is partly a consequence of differential sensitivity of the various proxies[16,18], but also because the changes in regional hydroclimates were modified by vegetation feedbacks[19] and local groundwater conditions[10]. A coherent spatial picture of the end of the AHP is emerging, as demonstrated in a recent synthesis of hydrological reconstructions[3], revealing a time transgressive termination of humid conditions from north to south (Fig. 2a).

Human occupation during the humid period is clearly demonstrated in numerous rock engravings and occupation sites, bearing evidence for the development of food production strategies and increasing socio-economic complexity[20,21]. Knowledge about spread and intensity of that human occupation is harder to acquire, yet enough exists to create a demographic reconstruction[22] (see Methods). Several major phases of population expansion and contraction can be identified in the Holocene Sahara from archaeological evidence. Hunter-Gatherer-Fisherfolk[21] initially colonised all regions around 10.5 ka with population levels peaking between 8 and 7.5 ka (Fig. 2b). Over the following millennium, northern Africa underwent a population decline, driven by a millennium-long dry event at 8 ka[9,16]. After 7 ka, domestic cattle, sheep and goat spread throughout northern Africa. This widespread adoption of (at least some) pastoralist strategies is followed by a second population boom (Fig. 2b). The second pulse of northern African human occupancy lasted until 5.5 ka, at which point the Sahara underwent a major population collapse, coinciding with the decline in favourable climatic conditions (Fig. 2). But was this climate–human interaction one way —or was the collapse of the green Sahara an early example of humans interfering with a sensitive environmental system?

We submit the suggestion that Humans were the catalyst of the collapse of the green Sahara[2] to a rigorous quantitative assessment. We first investigate whether the termination of the African Humid Period occurred early than expected, both through analysis of observations and using a model. We then examine the nature of early African pastoralism and its interactions with landscape. We conclude that the increased adoption of pastoralism provided a successful adaptation to the desertification caused by climate change.

## Results

**Natural length of the Holocene African Humid Period.** Before considering human agents in the context of climatic change, it is first necessary to determine the length of the Holocene AHP assuming no anthropogenic influence. Observations alone do not provide sufficient constraint on this, because of insufficiently accurate relevant chronologies. Mediterranean sapropel deposition is used as an indicator of humid conditions in northern Africa[23], because they have some of the most accurate chronologies. Over the past 250,000 years, it is possible to tune a chronology using well-dated speleothems to provide well-constrained estimates of the onset and termination of sapropels[24]. This chronology suggests that the most recent sapropel was of much shorter duration than previous instances (Fig. 3a); yet it only includes one other interglacial sapropel (at 129.5 ka). However longer records[25] that allow selection of similar orbital configurations[26] cannot detect differences at the sub-millennial timescales required (Fig. 3b). A concerted effort would be

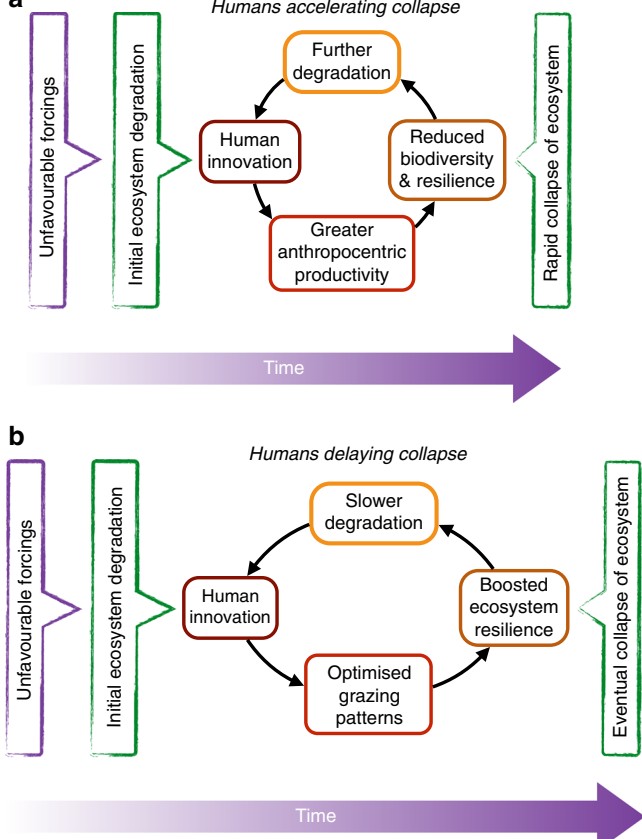

**Fig. 1** Pastoralist–environment interactions. **a** Schematic of a human population expansion beyond the carrying capacity of the region exacerbating aridification[2]. **b** Schematic of how the technological and cultural advances associated with sustainable pastoralism could help buffer changes to a fragile ecosystem

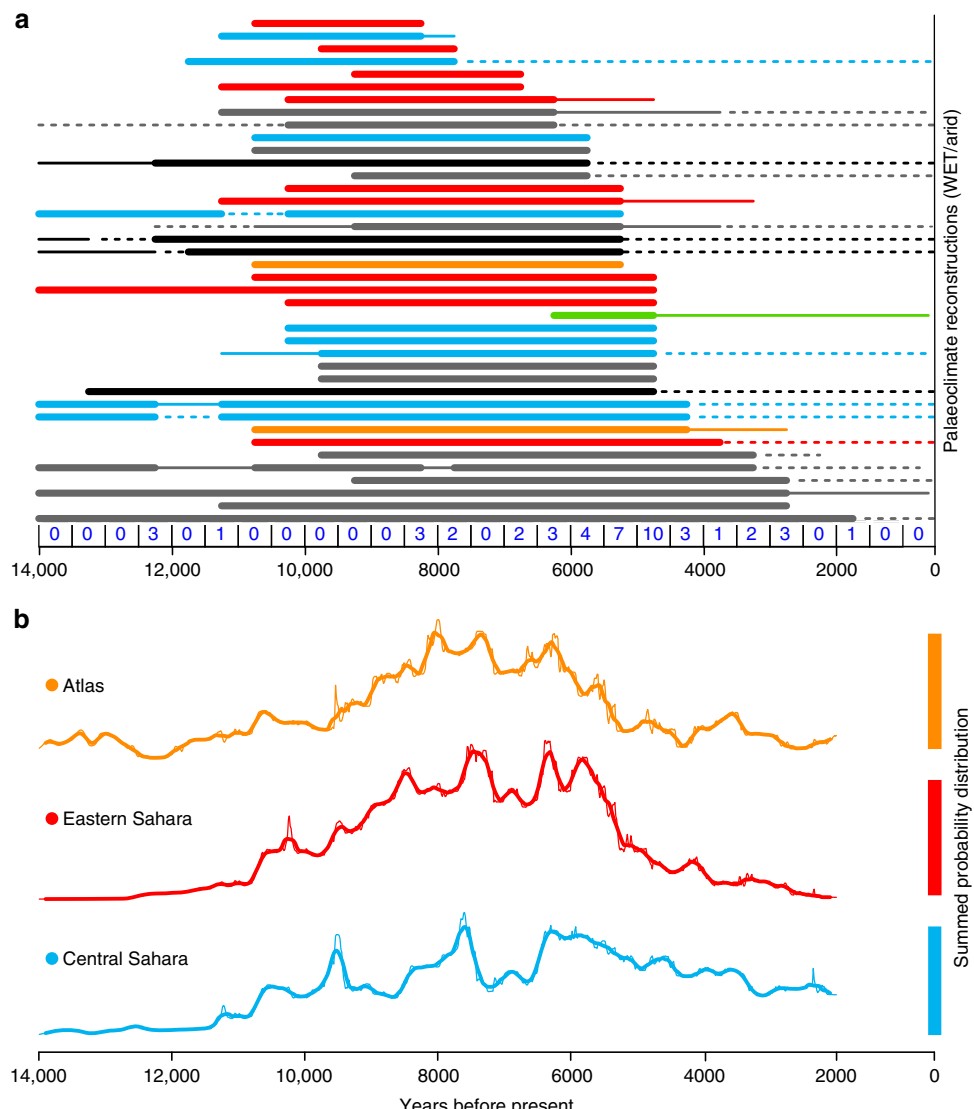

**Fig. 2** Reconstructions of Holocene northern Africa. **a** Palaeoclimate reconstructions[3] showing the existence of wet conditions (thick solid lines), semi-arid conditions (thin solid) or arid conditions (dashed). The individual reconstructions are colored by region: Central Sahara (Blue), Eastern Sahara (Red), the Atlas & Hoggar Mountains (Orange). The records discussed in the text are Lake Yoa[17] (green) and marine cores[13,14] (black), whilst records outside of the population regions are coloured gray. The records shown in order of their wet/dry transition, and the total number of wet/dry transitions is shown for each 500 year period. **b** Probability density of dated archaeological finds for the three regions in northern Africa, which can be used to infer the relative populations[22]

required to develop a sufficiently accurate chronology to advance in this direction.

We develop an idealised model that calculates rainfall and vegetative cover and their feedbacks (see Methods) to estimate the natural length of the Holocene AHP instead. Compared to previous models[27,28], rainfall responds to imposed orbital precession[29] and past greenhouse gas levels as measured in ice cores[30] (which acts as a proxy for glacial-interglacial changes as well as a local, direct forcing). The model is run over the past two glacial cycles (230–20 ka) using a large ensemble of parameter settings selected at random. Parameter settings that do not exhibit six green episodes during this period are discounted for being inconsistent with the observations. The remaining ensemble members are integrated forward to the present-day (Fig. 4). We find late Pleistocene behaviour alone was not sufficient to rule out the continuation of the humid period throughout the Holocene at the 5% significance level (Fig. 4). This failure to

accurately predict the passing of a known tipping point—despite having 200,000 years of observations—should add a cautionary note to the discussion surrounding future climate thresholds.

A sensitivity metric is devised for the model (see Methods) to summarise its behaviour and estimate start and end dates for the humid periods. We predict a well-defined start of the Holocene AHP (Supplementary Fig. 1), which corresponds closely with the observed date of 14.5 ka[9,10], supporting the validity of this modelling approach. The model shows several peaks during the Holocene when northern Africa would have been particularly sensitive to a perturbation (Fig. 5).

The largest peak in the modelled sensitivity of the Sahara occurs at 7–6 ka (Fig. 5c). This coincides with the second period of population increase between 6.7 and 6.3 ka (Fig. 5b). The dominant collapse observed for the Holocene AHP (Fig. 5a) occurs 500–1000 years after this peak (Fig. 5c), which appears to be a robust model result (see Methods, Supplementary Fig. 2).

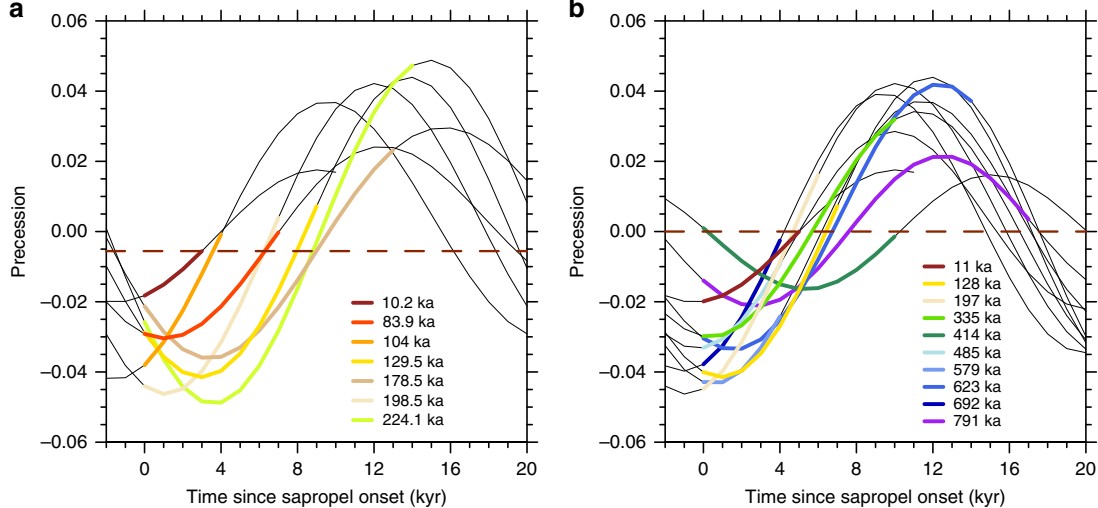

**Fig. 3** The relationship between sapropel formation and orbital precession using two sapropel chronologies. Precessional curves associated with sapropel formation are aligned to the start of each occurrence according to the respective chronology. The coloured segments of these curves indicate the actual duration of the sapropel. The red horizontal line indicates the precession at the termination of the most recent sapropel. **a** A speleothem-tuned chronology[24] provides well-constrained estimates of the onset and termination of sapropels over the past 250,000 years. **b** The past ten interglacial sapropels seen in a Mediterranean Sea level record[25]

This refutes the hypothesis that pastoralists were "active agents in landscape denudation" and accelerated the termination of the Holocene AHP[2]. Rather it suggests that pastoralism may have actively delayed the region's environmental deterioration (Fig. 1b).

**Robustness of the delay**. The synthesis of observed records[3] classifies the hydroclimate state only at 500 year intervals. This choice of interval was motivated by all the chronologies being sufficiently precise to resolve it[3]. The model inputs are orbital parameters[29] and carbon dioxide concentrations[30], both of which have dating uncertainties substantially less than 500 years. Dating of prior humid periods is subject to errors on the order of millennia (hence the failure to constrain the AHP dates observationally). Because of this issue, the valid model parameter settings are determined by matching solely the number of prior instances rather than their timing (see Methods). We consider the possibility that either a humid period was overlooked or that a sapropel has been laid down without a humid period during the past 230 kyrs to be minimal. The uncertainty contained within the structure of the idealised model, rather than its parameters, is impossible to quantify. To explore the parameter uncertainty in the model output, the whole experiment is replicated a further twenty times with different random parameter settings. There is little variation in the temporal structure (Supplementary Fig. 2). In summary, the limiting factor for the precision appears to be the temporal resolution of the compiled observations[3], though the delay appears visible despite that (Fig. 5).

The largest issues affecting the results of the idealised model are therefore associated with its applicability to the problem. There is a rich heritage of using idealised models to study the greening of the Sahara[27,28,31], so the application here is not without precedent. The model appears to adequately capture the past behaviour under certain parameter settings. We cannot exclude the possibility that including other natural forcing factors may be beneficial. An alternate approach would use coupled general circulation models (GCMs). These GCMs are now used operationally for decadal climate predictions[32]. Unfortunately,

the resources needed for the multi-millennia ensembles that would be required by this research preclude their application. Additionally, GCMs have been shown to have longstanding biases in simulating the greening of the Sahara[33], likely arising from them poor capturing of vegetation and dust feedbacks[34].

The model ensemble is treated above as multiple plausible instances of a single physical system. The sensitivity is therefore interpreted as a single metric for all three regions shown in Fig. 2. An alternate interpretation is that the ensemble members represent different local conditions, implying that the three sensitivity peaks in Fig. 5c each characterise a particular region. However, there is no noticeable regional pattern in the reconstructed collapse dates (Fig. 2a), although more southerly locations in the compilation do show a later response[3]. However, the majority of observational records showing a collapse between 6 and 5 ka[3] occur at similar latitudes to the archaeological sites used to estimate the human occupancy[22]. Therefore the comparison of the sensitivity metric to the palaeoclimate and population reconstructions combined across northern Africa seems appropriate (Fig. 5).

**Human–environment interactions**. The model results suggest that the end of the Holocene AHP was delayed by around 500 years. A logical extension from the hypothesis of anthropogenically-driven early collapse[2] is that humans caused this delay. Whilst other possible explanations could exist, the main difference between the Holocene and previous interglacials is the existence of Human society in the Holocene. We therefore explore whether mechanisms exist that may explain an anthropogenic role in the collapse, by focusing on why pastoralism is sustainable. This approach rejects any dualist view that humans occupy a unique place in nature[35], advocating instead the historical dependencies between human action and environmental change[36].

Mobility, a distinguishing feature of traditional pastoral systems[37], results in periodicity of the intensity of grazing. Grasslands can suffer from undergrazing as much as overgrazing[4,6], so active management of grazing plays a major role in grassland health. This is because grazing ungulates and

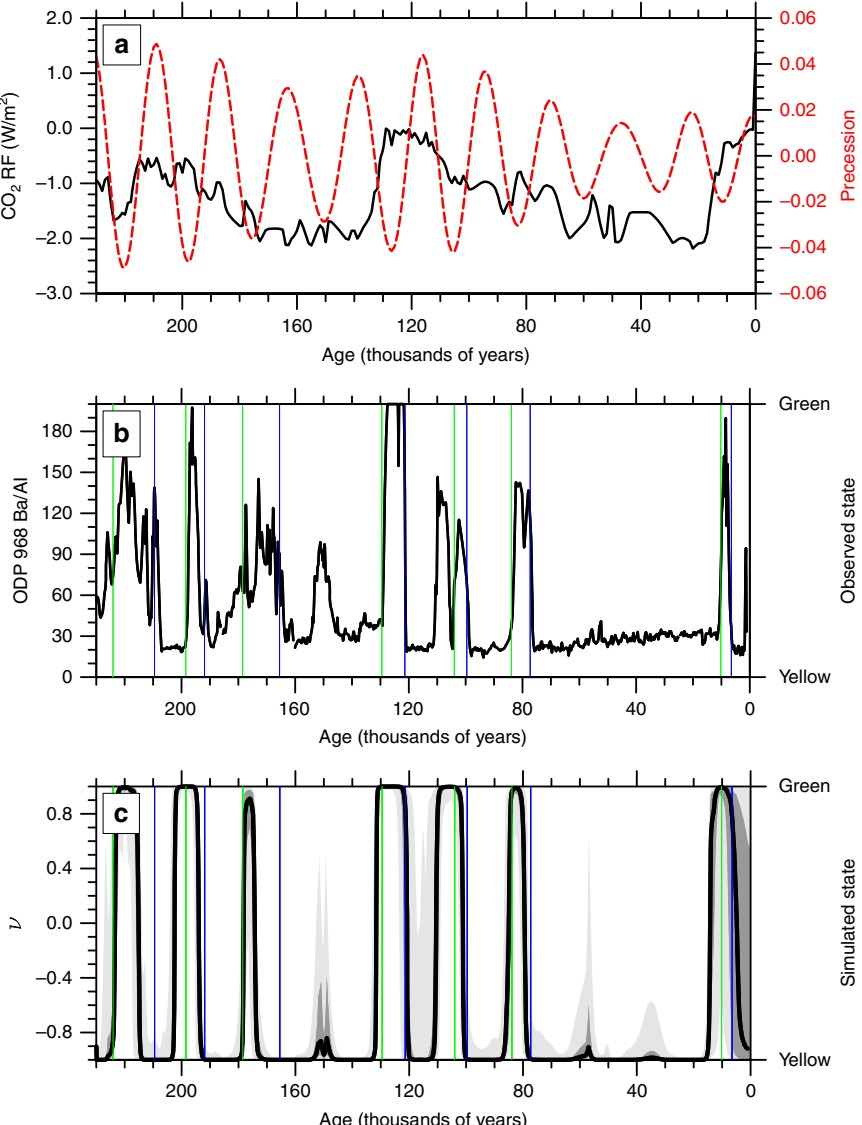

**Fig. 4** The last two glacial cycles. **a** The input times series of radiative forcing of carbon dioxide changes[30] (black) and climatic precession[29] (red). **b** Barium to aluminium ratio at Ocean Drilling Program site 968 in the Eastern Mediterranean[24]. **c** The distribution of the roughly 12,000 ensemble members that exhibit seven 'green' events. The median (black), inter-quartile range (dark gray) and 5–95% range (light gray) are shown, along with the sapropel start (green) and end (blue) dates calculated from observations[24]

grasslands have co-evolved from an historical predator-prey relationship, with pack hunting predators keeping large herds of ungulates bunched and moving[38]. Healthy grasslands are maintained in precisely this way by pastoralists bunching stock and moving them frequently, fostering a mutually beneficial distribution of dung and urine[37]. Removing grazers from grasslands increases the amount of senescent vegetation, which causes the grasses to cease growing productively[39]. Grazing livestock and their preference for the most palatable grasses provide a competitive advantage to the less palatable grasses for water and nutrients, making it important to get the balance correct between overgrazing and over-resting. Traditional pastoralists tend to be acutely aware of these subtle dynamics utilising practices that maximise grassland regeneration[38,40].

Evidence from long-term studies on herding strategies has also helped to reveal the sensitive dynamic between drought, pasture availability, and herd size. Seasonal and long-term droughts, which are common in areas of pastoral rangeland, as well as disease dynamics, control the growth of herds in a way that means they are unlikely to damage pasture. If longer-term drought starts to restrict pasture, or if herd size increases beyond the carrying capacity of a rangeland, then pastoralists will move on. For example, field research in the Ngorongoro Conservation Area has shown that whilst pastures were being overgrazed in terms of optimal commercial yield, this did not result in environmental degradation[41]. This is important as it suggests that animal condition deteriorates before they are capable of having a seriously deleterious effect on the environment. The amount of pastoralism practiced by the Saharan occupants, and therefore the size of their herds, are unlikely to have reached such levels as to surpass carrying capacity. The inherent mobility and customary institutions employed by these populations generates a dynamic state of adaptation, which logically negates over-burdening pastoral rangeland[5].

A recent publication by Wright[2] in which mid-Holocene pastoralists are considered "catalysts in accelerating the pace of devegetation in the Sahara" provides an illustrative example of the outdated doctrine against pastoralists. Wright[2] uses historical

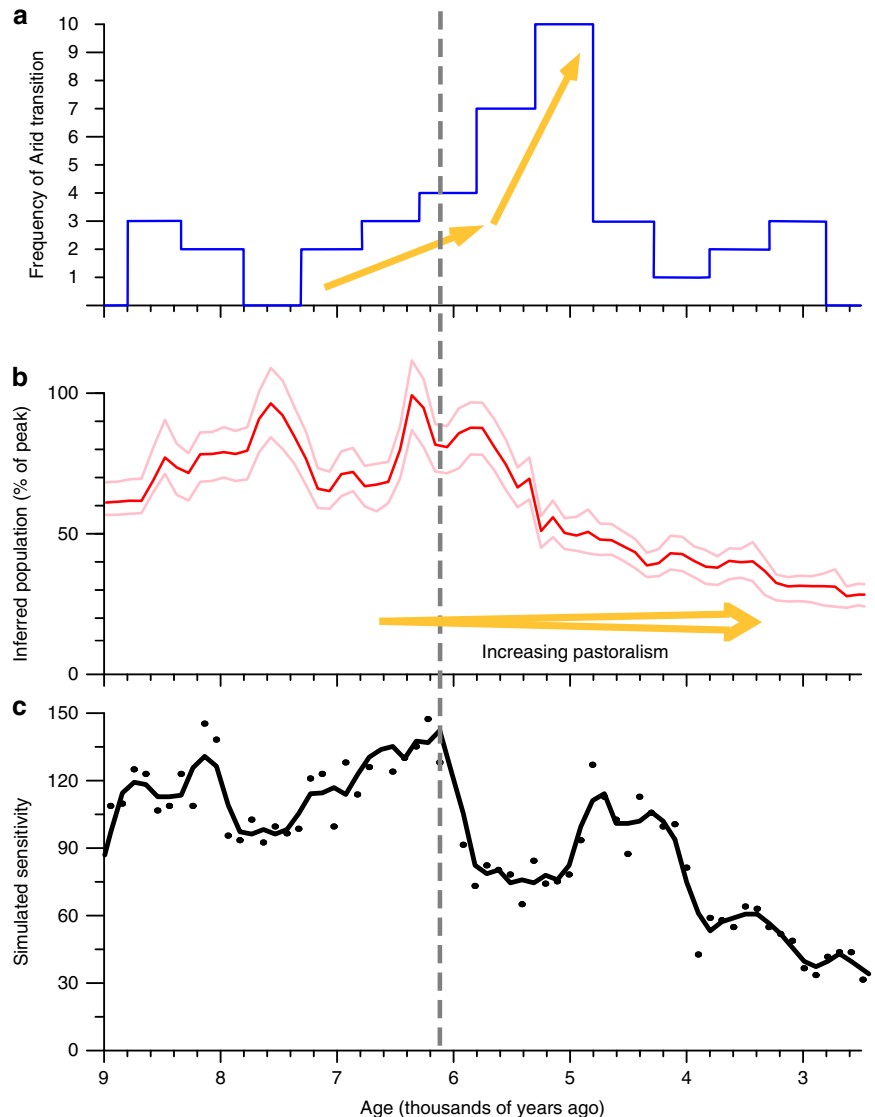

**Fig. 5** Potential interaction between humans and the ecosystem during the end of the African humid period. **a** Histogram of the number of climate proxies (Fig. 1a) indicating an end of the African Humid Period within a 500 year window[3]. **b** The population inferred summed probability distribution[22] over the whole of northern Africa region along with its 5–95% confidence level. **c** The simulated sensitivity of northern Africa diagnosed from the model. Black dots show the number of not-implausible model settings with a threshold time, $t^*$, in each century (see Methods for details); the black line is a 3 point running average. The gray line vertical indicates the time of the maximum simulated sensitivity

analogues, such as the Rapa Nui environmental degradation, that appear inappropriate. In the instance of the island of Rapa Nui, its inhabitants were primarily farmers and fishermen, not pastoralists. Even so recent research suggests that major environmental degradation on Rapa Nui occurred only after European contact, and that pre-contact changes in land use were a result of environmental constraint, not degradation[42,43]. Using this type of analogue, one establishes a false premise i.e., where "landscapes with no previous exposure to grazing by domesticated animals have been documented as crossing ecological thresholds shortly after new grazing pressures were introduced"[2]. Northern Africa, however, was becoming a domesticated landscape from the early Holocene onwards (Fig. 2). Pastoralism co-evolved with dryland environments in a context where extant grazing ungulates were in abundance. Moreover recent genetic analyses of modern African cattle indicate considerable introgression from African aurochs, suggesting they underwent a hybridization with local wild stock[44]. The introduction of pastoralist strategies, therefore, were based

upon natural ecosystem interactions and the functional roles of native wildlife causing little additional burden; allowing positive management of the environment.

**Regional responses**. The division of the entire Saharan population into broad regional sets (Fig. 2b) allows a preliminary look at spatial variation in the timing of population change. The population curves for the Eastern Sahara, the Atlas & Hoggar and Central Sahara start broadly synchronous; showing a rapid population increase after the onset of humid conditions c. 10.5 ka and during the millennial-long population decline between 7.5 and 6.5 ka (Fig. 2b). At the end of the AHP, however, we observe divergence in the regional demographic response. The eastern Sahara, which is today extremely arid, appears to have undergone a rapid population decline, as occupation shifted towards the Nile Valley. It has even been suggested that this subsequently gave rise to the Pharanoic civilisation[45]. To the north and west, in the Atlas & Hoggar mountain region, population decline appears to have

been equally rapid (c. 900 years, Fig. 2b). The central Sahara, on the other hand, saw a much more gradual decline in population levels that never reached the pre-Holocene population low (Fig. 2b). The fact that societies practicing pastoralism persisted in this region for so long, and invested both economically and ideologically in the local landscape, does not support a scenario of over-exploitation (see Methods). Additionally, the ethnographic record demonstrates how the flexibility inherent in traditional African pastoralist strategies enables them to make the most efficient use of patchy and fragile environments[4,5,37]. It is therefore likely that the origins of such strategies co-evolved with the drying environment in a way that enabled humans to live in an adaptive balance with available pasture.

The implication that Holocene populations persisted for longer in some parts of the Sahara either suggests a spatial variation in the rate of aridification or vegetation change, or more intriguingly in the human adaptive strategies. Differential topography across the Sahara is certainly worth considering. Mountains such as the Tibesti, Tassili-n-Ajjer and Ahaggar form a major topographic feature spanning more than 2500 km from southern Algeria to northern Chad. These mountains would have acted as important water towers in contrast to the surrounding plains, providing populations living on the windward side with more persistent rain runoff during periods of increasing aridity. Some of the earliest direct evidence for the exploitation of domestic livestock[46], use of milk products[47], and the construction of cattle tumuli[46,48], come from the heart of the central Sahara. On the Messak plateau, for example, extensive evidence for rock art depicting livestock scenes and stone monuments with associated domestic animal remains dating to the middle Holocene attest to a highly formalized expression of a wider Saharan "cattle cult"[46,48]. Isotopic analysis of archaeological animal bones from this region also demonstrate seasonal transhumance[48], reminiscent of the strategies used by modern traditional pastoralists to ensure the maintenance of healthy pasture.

## Discussion

The possibility that humans could have had a stabilising influence on the environment has significant implications. Naturally there are consequences for our understanding of past climate changes. For example, there is a long-standing discrepancy between observed climate of 6 ka for northern Africa and simulations by global climate models[33], which currently include no pastoralism. Also the "early Anthropocene" hypothesis[49] identifies a human-caused perturbation in the carbon cycle around the time of the aridification of northern Africa. It is doubtful that the anthropogenic delay suggested by the model results above could perturb the global carbon cycle. The carbon stored in northern Africa vegetation would have been relatively insignificant. One would need to invoke speculative, remote impacts on both tropical wetland methane emissions and the carbon sequestration in rainforest peatlands[50].

More broadly, this work presents a positive message about sustainability and climate adaptation. We contest the common narrative that past human–environment interactions must always be one of over-exploitation and degradation[51] (Fig. 1a). This study shows that increasing human population combined with an intensification of pastoralism did not accelerate aridification, and may even have delayed the collapse of the green Sahara (Fig. 1b). This finding provides yet more evidence for the sustainability of pastoralism[4]. It suggests that traditional, indigenous practices were developed as an adaptation to Holocene climate change in northern Africa. Promoting and enhancing sustainable pastoralism could be a vital adaptation to our current climate challenge.

## Methods

**Data**. Proxy records of northern African palaeoclimate are derived from a variety of sources. These range from lake-level, dust deposition, pollen and geochemical records. The data used in Figs. 2a, 5a are derived from the database compiled and interpreted by Shanahan et al.[3]. For every 500 year interval, the climate state has been subjectively determined[3] as either wet, moderate or dry (Fig. 2a). As with the sensitivity metric (Eq. 7), we date the collapse as the first time in which humid conditions are not present (Fig. 5). This compilation of proxy records may provide a geographically and typographically biased sample, but is not clear what alternate approaches are availabel to estimate an end-date for the green Sahara in a probabilistic fashion.

The relative population levels (Figs. 2b, 5b) are a summed probability distribution analysis based on a comprehensive review of the abundance of carbon-14 dated archaeological sites across northern Africa[22]. The underlying principle of this method assumes a monotonic relationship between the amount of data and the amount of human presence, which is reliant on the law of large numbers to overcome small-scale temporal and spatial biases. Full details on the methods are described in Shennan et al.[52], whilst criticisms[53,54], and subsequent defense[55,56] of the method have been presented in several publications. The population estimates used in the present analysis[22] were created from a dataset comprising 3287 radiocarbon dates from 1011 "Neolithic" sites. Radiocarbon dates from state level social contexts such as Pharaonic or later Garamantian sites were not included in that analysis. The population estimates can only provide relative time series and the size of populations cannot be compared between the regions shown in Fig. 2b. To date, these are the only explicit reconstructions of Holocene demographic trends on a trans-Saharan scale, although similar curves have been produced for the western desert in Egypt[45]. Furthermore, it is this curve which Wright suggests corresponds with "the variable tempo and intensity of the termination of the AHP" and "local transitions to shrubland environments and accelerated rates of soil erosion"[2]. We exclude African palaeoclimate reconstructions south of 13.42°N from our analysis, as this is the most southerly archaeological site used to reconstruct the population estimates[22].

**Idealised model formulation**. The simplest model of climate–vegetation interactions consists of the vegetation cover being determined by rainfall, which itself depends on external forcing and vegetation cover[31]. We adapt the non-dimensionalised model of Liu[28] that captures inter-annual variability[27] with the modification that the time-invariant background rainfall is now considered a linearised function of precession and carbon dioxide forcing. This idealised model incorporates a vegetation cover, $\nu$, that ranges from shrubland (1; "green") to desert ($-1$; "yellow"). The vegetation cover changes at a rate

$$\frac{d\nu}{dt} = \frac{1}{\tau_\nu}\tanh(R) - \nu, \tag{1}$$

where $\tau_\nu$ is the vegetation timescale (in years) and $R$ is the non-dimensionalised rainfall. $R$ is centred around a sensitive range that spreads from $(-1, 1)$. It is given by:

$$R = a + bP + cF + d\nu + N, \tag{2}$$

where $P$ is the eccentricity-modulated precession[29], $\varepsilon\sin\bar{\omega}$, (Fig. 4a) and $F$ is the radiative forcing with respect to the preindustrial. Here the radiative forcing (Fig. 4a) represents solely carbon-di-oxide and is calculated as 5.35 $\ln(CO_2/278)$, where $CO_2$ is the carbon dioxide concentration[30] in parts-per-million by volume [278 ppm was the preindustrial concentration]. The feedback of vegetation onto the rainfall is captured by the $d\nu$ term in Eq. 2, where $d$ sets the magnitude of the feedback. Previous work[28] has used $d$ ranging from 0.8 to 1.2; a wider range is sampled here to encompass a broader spread of uncertainty (see Supplementary Table 1). The red noise term, $N$, is given by

$$\frac{dN}{dt} = \frac{\sigma\varsigma(t) - N}{\tau_N}, \tag{3}$$

where $\tau_N$ is soil moisture timescale (in years) and $\varsigma$ is a random sample from a unit normal distribution scaled by a tunable parameter, $\sigma$.

The impact of a doubling in $CO_2$ has previously been shown to expand the critical range of rainfall[31]. However, it is incorporated here as an additive term (expressed as a radiative forcing change from preindustrial in W/m²) as attempts with a multiplicative factor were unsuccessful in replicating the observed lack of green states during MIS3 (Fig. 4b). The modified background rainfall, $a + bP + cF$, must at times be less than 1 otherwise the system would never leave the green state and is generally less than 0 to prevent the green state becoming the predominant condition.

Iteration is achieved through a forward timestepping approach[28] with a timestep, $\Delta t$, of 1 year.

$$\nu_{k+1} = \nu_k + \frac{\Delta t}{\tau_\nu}\left[\tanh\left(\frac{a+bP}{cCO_2} + d\nu + N_k\right) - \nu_k\right], \quad (4)$$

$$N_{k+1} = N_k - \frac{N_k \Delta t}{\tau_N} + \frac{\sqrt{\Delta t}\sigma W_k}{\tau_N} \quad (5)$$

Previous work[28] has shown that this system can exhibit bimodality (switching between two different states) despite being monostable (i.e., having a single potential well, Eq. 6). The stochasticity (Eq. 3, best thought of as interannual variability in the soil moisture[27,28]) combined with the non-linear dependence of vegetation on rainfall (Eq. 1) can lead to the simulation often passing through the state with minimum equilibrium potential[28]. The bimodality explored previously in this style of system[28] occurs with a background rainfall (and hence minimum equilibrium potential) centred on $\nu = 0$. It is under this condition that the system is most responsive to noise. Otherwise (as for the vast majority of the 230 ka simulated here), the stochastic contribution is effectively biased towards either the green or yellow state. This means the model is not exhibiting the canonical form of abrupt collapse (i.e., a bistable system rapidly flipping state). Rather this model represents forced changes overprinted with substantial stochasticity, which leads to shifts between two predominant states that may be abrupt in nature.

The idealised model has seven unknown parameters: three related to the background rainfall ($a$, $b$, and $c$); the feedback strength, $d$; two inherent timescales ($\tau_\nu$ and $\tau_N$); and the climate noise scaling, $\sigma$. These cannot be individually constrained from observations, in part due to their idealised nature. A 100,000-member ensemble is created to explore parameter and internal variability uncertainty. For each ensemble member, the values of the seven parameters are randomly selected from a uniform distribution over the ranges shown in Supplementary Table 1. The remaining subset of 12,099 simulations are considered as "not implausible". Interestingly roughly a third ($n = 3534$) of this subset never leave the green state during the Holocene.

In the absence of stochastic noise, the equilibrium potential for the idealised model above is

$$U(\nu) = \frac{\nu^2}{2} - \frac{\ln(\cosh(a+bP+cCO_2+d\nu))}{d} \quad (6)$$

**Simulated sensitivity metric**. If the noiseless system were left to reach equilibrium with a given forcing, it would end in the state with the minimum equilibrium potential. The time-varying nature of the forcings suggests that even with the addition of noise an individual model simulation can be adequately approximated by its equilibrium state (Supplementary Fig. 1A). This permits identification of when the system should flip between the green and yellow states. We define a threshold time, $t^*$, at which the minimum equilibrium potential changes side of the $\nu = 0$ line (Supplementary Fig. 1A). Following from Eq. 6, the threshold time, $t^*$, occurs when

$$\mathrm{sgn}(a+bP_{t^*}+cF_{t^*}) \neq \mathrm{sgn}\left(a+bP_{(t^*-\Delta t)}+cF_{(t^*-\Delta t)}\right) \quad (7)$$

This allows us to define a simulated sensitivity metric as the number of not-implausible ensemble members exhibiting threshold behaviour at that time, i.e., SS $(t) = n(t^* = t)$. Exclusion of ensemble members that do not collapse during the Holocene does not alter the simulated sensitivity time series. This simulated sensitivity shows a definite spike at 14.7 ka (Supplementary Fig. 1B) demonstrating the ability of our approach to capture the onset of the African humid period. Such a consistent signal is not shown for its termination (Fig. 5c).

An alternate approach to sampling the uncertainty contained within the model's tunable parameters would be to only select the ensemble members with a good fit to observations. Selecting just the 1500 ensemble members best correlated (i.e., with the highest $R^2$ values) to the Ba/Al observations shown in Fig. 4b would lead to a single sole peak in simulated sensitivity at ~6.5 ka. Given that sapropel S1 is observed to terminate ~1000 years earlier than the compilation in Fig. 5a[3,24], it would be hard to conclude an anthropogenic delay from this subset of best-correlated models[57]. Our "not implausible" approach is only conditioned on sapropel existence rather than timing—removing any circularity.

**Code availability**. The idealised model has been programmed in NCL, as were all the codes to plot the figures presented here. A single model instance for the Holocene has been written in Python as a Jupyter Notebook. All programs can be accessed from the repository via the EarthArXiv at https://doi.org/10.17605/OSF.IO/WYAFZ.

## Data availability

Much of the data shown in this manuscript has been previously published elsewhere. Nonetheless all data shown in the individual figures can be accessed from the repository via the EarthArXiv at https://doi.org/10.17605/OSF.IO/WYAFZ.

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

## Acknowledgements

Martin Ziegler, Tim Shanahan and Eelco Rohling kindly provided data—along with advice on its use. Zhenghyu Liu gave timely and helpful advice during the model development. David Thornalley, Adrian Timpson, Jonathan Holmes, Chronis Tzedakis, Charlie Bristow, as well as Bill Ruddiman, joined in fruitful discussions. Katie Manning was supported by the Leverhulme Trust (RPG-2016-115) and the European Research Council project (249390).

## Author contributions

CB conceived the project with KM. The observational and modelling results were generated by CB. KM developed the discussion around pastoralist feedbacks with MM and CB. CB and MM developed and refined the diagrams. All authors contributed to the ideas and text contained in the manuscript.

## Additional information

**Competing interests:** The authors declare no competing interests.

