## [Peer Review File · Nature Communications]

Reviewers' comments:

Reviewer #1 (Remarks to the Author):

Pastoralists delayed the collapse of the green Sahara

Chris Brierley, Katie Manning, and Mark Maslin

This communication presents an alternative to the paper published by Wright (2017) claiming the advent of pastoralism contributed to the onset of the end of the African Humid Period. Wright's claims are problematic to say the least and it is appropriate that they are questioned here. Brierley and colleagues also present some very useful observations regarding the sustainability of pastoralism as an economic system in North Africa.

There are three major issues raised by this paper that are not sufficiently dealt with, and they largely relate to theorising human-environment interrelationships and the archaeological record of human occupation during the Holocene in North Africa. I believe this generally relates to a failure to consider recent archaeological and anthropological discussion on the subject of human environment interrelationships for North Africa and in general (for example see Bauer and Ellis 2018 for discussion on the Anthropocene).

1.

Prime mover arguments were popular in archaeology during the 20th century, particularly related to key changes in subsistence strategy during the Holocene. Correlations between climate and socio-economy have been common in North Africa, and the use of the African Humid Period in particular to explain everything from cattle domestication to the rise of pharaonic civilisation in Egypt permeates the literature. While pastoralism is certainly an important economic solution in some marginal, arid environments, the relationships between people and their environments, and their historically contingent and varied socio-economies is extremely complex. Simple models, such as the one presented here are powerful, but the authors need to consider this potential complexity. Perhaps including some discussion on the theoretical frameworks used to understand human-environment interrelationships would be useful.

2.

My first point also relates to the characterisation of socio-economy. Creating a dichotomy (as in figure 4) between pastoralists and h/g may be a little premature for this phase of the Holocene. Archaeological evidence does not support either a rapid or complete transition, rather the persistence of a broad spectrum of resource use, probably more appropriately considered 'low level food production' (sensu Smith 2001). In fact many of the archaeological sites from which the radio-carbon corpus is drawn support evidence for both strategies, and in some cases very little evidence for pastoralism is found at all (see Holdaway and Phillipps 2017 for summary). Modern or historic ethnographic evidence, while useful, may obscure potential variability in the past. It cannot be assumed the same systems were in play during the mid-Holocene, or indeed characterise the entire region based on a few examples. Again some discussion of this complexity and acknowledgement of the inherent over-simplification of the model is necessary.

3.

The third issue relates to the population modelling from the existing radiocarbon chronology, based on data published by Manning and Timpson (2014). I appreciate the intent of this communication is not to describe the population model, but it is critical to addressing the issue of causation and the ability to 'refute the hypothesis' tested here.

Human demography is challenging; it can be tempting to use radiocarbon determinations as a proxy for population, both occupation intensity and as a gross estimate of numbers. However, many studies (e.g. Davies et al. 2015) point out issues in the samples to begin with. To my knowledge there has been no systematic critical assessment of the radiocarbon corpus from North Africa, with the exception of a graduate thesis (Phillips 2013). This preliminary work reveals inherent biases in the dataset, concentrations of radiocarbon dates are largely a function of where archaeologists have looked, the number of years a project runs, and presumably budget. This is

even before we consider other issues with the dataset, such as those raised by Vermeersch (2014). This is obviously critical to the assumptions made of a population expansion associated with pastoralism. One article is referenced to explain the population increase, but this needs to be dealt with – how are population fluctuations, so critical to the argument, actually empirically demonstrated? A ‘dots on map’ approach assuming a relationship between population size and the amount of radiocarbon ‘dates’ is not sufficient in light of recent critiques. The methods section describes the use of summed probability plots from radiocarbon determinations from across North Africa. No information is given on the sampling strategy employed to procure this dataset and this is not sufficiently dealt with in Manning and Timpson (2014). At the very least, some discussion of the critical issues of this analysis should be discussed. In addition more radiocarbon determinations have been published for the region since the 2014 work. Population migration from the eastern Sahara is then suggested have given rise to Pharaonic civilisation; again the radiocarbon chronology upon which this is based is problematic, and recent consideration of archaeological evidence does not adequately support this hypothesis (Wengrow et al. 2014).

Many archaeologists based in North Africa have worked very hard to demonstrate the complexity of human behavioural response to, and impact on, the environment. At this point in the 21st century it would seem more productive to advance theoretical frameworks rather than return to old ones.

Texts cited:

Wright, DK. 2017. Humans as Agents in the Termination of the African Humid Period. *Front. Earth Sci.* 5:4. doi: 10.3389/feart.2017.00004

Bauer, A. M. and Ellis, E. C. The Anthropocene Divide: Obscuring Understanding of Social-Environmental Change. *Current Anthropology* 59.2 (2018): 209-227.

Smith, Bruce D. 2001. Low-level food production. *Journal of Archaeological Research* 9.1: 1-43.

Holdaway, S., and R. Phillipps. 2017. Human environmental interrelationships and the origins of agriculture in Egypt and Sudan. *Oxford Research Encyclopedia of Environmental Science*.

Manning, K. and A. Timpson. 2014. The demographic response to Holocene climate change in the Sahara. *Quaternary Science Reviews* 101: 28-35.

Davies, B, S. J. Holdaway, and P. C. Fanning. 2016. Modelling the palimpsest: an exploratory agent-based model of surface archaeological deposit formation in a fluvial arid Australian landscape. *The Holocene* 26.3 : 450-463.

Phillips, N, 2013. Assessing the Temporal Foundations of Supra-Regional Models for Early to Mid-Holocene Climate-Cultural Change, Northeast Africa. MA Thesis, University of Auckland.

Vermeersch, P. 2014. Comment on “The demographic response to Holocene climate change in the Sahara”, by Katie Manning and Adrian Timpson. *Quaternary Science Reviews* 110 (2015): 172-175.

Wengrow, D, et al. Cultural convergence in the Neolithic of the Nile Valley: a prehistoric perspective on Egypt's place in Africa. *Antiquity* 88.339 (2014): 95-111.

Reviewer #2 (Remarks to the Author):

In this study the authors present a modeling study to investigate whether the development of pastoralist societies in Northern Africa during the early and mid Holocene may have had any role in the desertification of the Sahara region at the end of the African Humid period around 5500 years ago. The main conclusion of the work is that the pastoralism may have slowed down the

deterioration caused by orbitally-driven climate change rather than enhanced as suggested in Wright et al. (2017).

This is an important message that have broad implications and I totally agree with the authors when stating that Wright et al. provide “an illustrative example of the outdated doctrine against pastoralists”. I actually wonder why Wright et al study received so much attention to even be reported by BBC mundo (<http://www.bbc.com/mundo/noticias-39307995>). It is indeed well known that light-to-moderate grazing actually favors plant species biodiversity, while heavy grazing livestock production can be detrimental and no grazing may also cause it to plummet (Hart, 2001).

My major concern with the present study regards the fact whether the simplified model used by the authors can be used to draw such conclusion, i.e. that the Holocene green-Sahara period may have actually ended earlier without the pastoralism. As the authors also state, more complex climate models still struggle to replicate the Sahara greening during the African Humid Periods, nevertheless I am not fully confident that such a simplified model can be used for the specific question the authors are trying to answer. For example while the model does a good job in replicating the wet and dry periods it is not always in agreement with the sapropel start and end.

Minor comments

1) When submit a manuscript for review is always useful to have the line numbers.

First paragraph:

2) Please change the verb forecast with either simulate or predict

Main text:

3) Together with reference 11 and 12 and I suggest to add Hart, 2001.

4) Please clarify this sentence: “The model shows several times during the Holocene when northern Africa would have been particularly sensitive to interference (Fig. 4).

5) Please clarify and fix this sentence: “Mobility, a distinguishing feature of traditional pastoral systems, promotes periodicity and in the intensity of grazing, which plays a major role in grassland health as grasslands can suffer from undergrazing as much as overgrazing¹¹”. Add also Hart 2001.

References:

6) Fix reference 8

7) Is reference 36 still a Discussion paper? Please update it.

Methods:

8) Regarding: “The ‘natural’ length of the present interglacial has been estimated from observations³⁶ and potential anthropogenic perturbations highlighted within it²⁸.” The authors do refer to a study, but it would be helpful to have a short summary of how the natural length of the present interglacial has been estimated.

The following sentence is also not clear. Why ref. 28 could estimate the natural length of the present interglacial and the authors cannot?

9) Change “model of Lui⁸” to read “model of Liu⁸”

10) Fix: “The time-varying nature of the forcings mean”.

11) Regarding: “Given that sapropel S1 is observed to terminate earlier than the compilation in Fig. 4 A”. It would be good to specify here how much earlier.

12) Regarding: “It is worth noting that models which are more complicated are not able to adequately simulate the greening of the Sahara”. I would suggest to spend few extra lines in presenting why the CMIP/PMIP models most likely fail in representing the Green Sahara (most likely lack of vegetation and dust feedbacks). I would avoid the use of “It is worth noting” and similar expressions, since if it were not worth noting something, the authors would not write about it.

13) The last two sections “Central Saharan pastoral persistence” and “Misunderstandings propagated by Wright” should be included in the main text.

Acknowledgments:

14) Please correct the name of Zhengyu Liu.

Figures:

15) Figure 1: What is shown in y-axis in both panels? Missing the labels

16) Figure 4 missing numbers on the y-axes of panel A and B.

References

1. Hart, 2001: Plant biodiversity on shortgrass steppe after 55 years of zero, light, moderate, or heavy cattle grazing, *Plant Ecology*, Volume 155, Issue 1, pp 111–118
<https://link.springer.com/article/10.1023%2FA%3A1013273400543>

Reviewer #3 (Remarks to the Author):

Report on

'Pastoralists delayed the collapse of the green Sahara'

by

Bierley, Manning, and Maslin

The authors tackle a very interesting topic: the interaction of pastoralism and ecosystem dynamics in semi-arid regions. They propose that pastoralism could be beneficial for marginal, arid environments, and they list a number of plausible mechanisms to support their proposition. This sounds convincing. However, when put into context with the changes in Holocene Saharan vegetation, the line of arguments appears to be less convincing.

The title of the paper sounds pretty catchy, addressing the 'collapse of the green Sahara'. The rapid increase in dust flux from the Sahara into the North Atlantic shelf as seen in the reconstructions by deMenocal et al (2000) and Mc Gee (2013) seems to support the earlier prediction of a tipping point, or collapse, by Brovkin et al. (1998) and Claussen et al. (1999). If one looks at the dust records, however, the increase in dust flux is perhaps an order of magnitude faster than the change in orbital forcing. Hence, geologically speaking, this might be considered a collapse, but a collapse that took many centuries or several dozen generations of humans that once lived in the Holocene Sahara. They may have noticed a very slow increase in aridity, perhaps mainly from hearsay by their ancestors. Figure 2B of this paper supports the very slow changes in the population of the green Sahara specifically around 5,500 years before present, when the rapid change in the Holocene Sahara occurred. Hence, the term "collapse of the green Sahara" in relation to human dynamics in the Holocene Sahara is surely a bit overstated.

The authors support their proposition by showing that their model predicts a somewhat earlier possible termination of the African Humid period than reconstructed. The prediction is internally consistent, the choice of model parameters and the estimate of the robustness of model simulations are also convincing. However, the conclusion that human activity must be the cause for the difference between predicted end of the AHP and number of climate proxies indicating the end the AHP is not compelling. This difference could be attributed to a number of other factors.

The model used by the authors is fascinating, and it is a great tool for conceptual visualisation. But does it capture the complex dynamics and spatial heterogeneity of the Holocene Sahara? Does it really allow to 'forecast the length of the AHP'? A number of studies, reconstruction (e.g. Shanahan et al., 2015) and modelling (Bathiany et al., 2012, Egerer et al., 2018), have demonstrated that rapid changes in the Holocene Sahara can occur in different regions at different

times in the past. Possibly, fast transitions prevailed more in the Western part, and more gradual transitions, in the Eastern part (Brovkin and Claussen, 2008). This complexity is not captured by the conceptual model.

In line with the previous argument: Is the number of proxies indicating the end of the AHP really a proper measure of the end of the AHP? According to the studies mentioned above, the increasing number of indicators for the end of the AHP does not indicate an increased probability of the end of the AHP as a whole, but rather reflects the time-transgressive shift of the AHP termination. Aggregation of geographical heterogeneity is perhaps the weakest argument of this study.

I would like to conclude by saying that I like the study as it critically re-assesses and disputes the conventional idea of degradation due to overgrazing. The proposition outlined by the authors might even be valid for the Holocene Sahara. But it is only a hypothesis, not a compelling proof or prediction. Therefore I suggest that this study be published only after careful re-writing and stressing the hypothetical character of the modelling. For example, phrasing the title as a question rather than a seemingly affirmative statement would be one item. Consistently, the hypothetical character of the modelling, specifically its limitation regarding geographical complexity, should be clearly stressed throughout the text.

Minor comments:

2nd paragraph: The rate of change of precipitation (drying) and in vegetation (desertification) do not necessarily go hand in hand. Liu et al. (2006) differentiate between "stable" and "unstable" collapse which emerge from a loss of stability of just from a fast feedback, respectively. However, this only applies in a statistical sense (Claussen et al., 2013). A single realisation, or record such as the Lake Yoa record, which reveals a gradual decline in rainfall, does not necessarily refute the existence of a bifurcation.

2nd paragraph: The term "green Sahara" was surely not coined by Lenton et al. (2008). The Hungarian adventurer László Almásy who found the cave of swimmers the Gilf Kebir wrote about the green Sahara in his book 'The unknown Sahara' in the 1930s. This term was used later in a number of palaeo climate papers in the 1990s.

3rd paragraph: The authors stress that their model – in contrast to the model by Liu et al. (2006/7) – includes the effect of CO₂ on rainfall. This effect is, of course, a rather indirect one. CO₂ fertilisation might also play a role. It seems that changes in CO₂ are important for the tuning of the conceptual model. But for the green Sahara problem it might be rather unimportant.

6th paragraph (or where appropriate): What about fire ecology? People have used fire, and presumably, herders have used fire in a different way and intensity than early farmers did.

Penultimate paragraph: I do not understand the last sentence. Which teleconnection is meant? Perhaps, the link between Holocene pastoralists in the Sahara and the global carbon cycle is a bit far-fetched.

At a number of places, the authors emphasise that global, comprehensive models do not properly describe the Holocene greening of the Sahara. Indeed, most comprehensive models yield a small greening (although the first 'green model' (Claussen and Gayler, 1997) shows a substantial greening). However, this statement does not imply that the conceptual model of the 'Brovkin/Liu type' is superior. Both types of models serve different purposes.

Literature (not included in the paper)

Bathiany et al. (2012). Implications of climate variability for the detection of multiple equilibria and for rapid transitions in the atmosphere-vegetation system. *Climate Dynamics*, 38 (9-10), 1775-1790.

Brovkin and Claussen (2008). Comment on "Climate-Driven Ecosystem Succession in the Sahara: The Past 6000 Years. *Science*, 322, 1326b.

Claussen et al. (1999). Simulation of an abrupt change in Saharan vegetation at the end of the mid-Holocene. *Geophysical Research Letters*, 24 (14), 2037-2040.

Claussen et al. (2013). Simulated climate-vegetation interaction in semi-arid regions affected by plant diversity. *Nature Geoscience*, 6 (11), 954-958.

Egerer et al. (2018). Rapid increase in simulated North Atlantic dust deposition due to fast change of northwest African landscape during the Holocene. *Clim. Past Discuss.*, <https://doi.org/10.5194/cp-2018-39>, in review, 2018.

1 Overall Comments

We appreciate the diligence and effort devoted by the reviewers to assess our manuscript. There are many specific comments, which we respond to individually below. Overall though there was a sense across the three reviewers that we were being more assertive that the evidence warranted. We acknowledge that more convincing research could be performed with a serious, dedicated interdisciplinary effort. Nonetheless, there is also a strong possibility that sufficient proof may never exist. Instead we have focused on trying to join disciplines together at their current status.

We hope with this manuscript to open up a discussion around the potential role of early humans in the Earth System, by posing an interesting question. As the reviewers rightly state, there are many elements where further research would constrain some of the uncertainties. We envisage this manuscript will motivate that research. We feel that we had alluded to most of the caveats in either the main text or the methods of our previous version. In this version, we have strengthened the discussion of our research's limitations and moved text from methods into the main text. We hope that this greater openness means that the text now better describes the true state of discussion. We have additionally adopted a less forthright title.

2 Reviewer #1

This communication presents an alternative to the paper published by Wright (2017) claiming the advent of pastoralism contributed to the onset of the end of the African Humid Period. Wright's claims are problematic to say the least and it is appropriate that they are questioned here. Brierley and colleagues also present some very useful observations regarding the sustainability of pastoralism as an economic system in North Africa.

- Thank you for this kind summary.

There are three major issues raised by this paper that are not sufficiently dealt with, and they largely relate to theorising human-environment interrelationships and the archaeological record of human occupation during the Holocene in North Africa. I believe this generally relates to a failure to consider recent archaeological and anthropological discussion on the subject of human environment interrelationships for North Africa and in general (for example, see Bauer and Ellis 2018 for discussion on the Anthropocene).

- We recognise that we have been somewhat brief with our discussion about the topic in this necessarily short letter. However, we would like to highlight that one of us (MM) has been actively working to include greater recognition of archaeology and anthropology into the Anthropocene's classification.

1. Prime mover arguments were popular in archaeology during the 20th century, particularly related to key changes in subsistence strategy during the Holocene. Correlations between climate and socio-economy have been common in North Africa, and the use of the African Humid Period in particular to explain everything from cattle domestication to the rise of pharaonic civilisation in Egypt permeates the literature. While pastoralism is certainly an important economic solution in some marginal, arid environments, the relationships between people and their environments, and their historically contingent and varied socio-economies is extremely complex. Simple models, such as the one presented here are powerful, but the authors need to consider this potential complexity. Perhaps including some discussion on the theoretical frameworks used to understand human-environment interrelationships would be useful.

- We had not felt that we had adopted an explicit theoretical framework, because by nature we are presenting a very simple, broad scale model. In our opinion, the premise of the modelling is not really to address the 'historically contingent and varied socio-economies'. Rather it is to explore simple, but powerful explanatory parameters. Importantly, we feel this is not in contradiction to historical contingencies, but simply operates at a broader scale of enquiry. Nonetheless, we must not convey that clearly in our original manuscript to the reviewer. We therefore now include a brief discussion of the placement of our research with respect to existing theoretical frameworks (L107-8 of the revised manuscript).

2. My first point also relates to the characterisation of socio-economy. Creating a dichotomy (as in figure 4) between pastoralists and h/g may be a little premature for this phase of the Holocene. Archaeological evidence does not support either a rapid or complete transition, rather the persistence of a broad spectrum of resource use, probably more appropriately considered 'low level food production' (sensu Smith 2001). In fact many of the archaeological sites from which the radio-carbon corpus is drawn support evidence for both strategies, and in some cases very little evidence for pastoralism is found at all (see Holdaway and Phillipps 2017 for summary). Modern or historic ethnographic evidence, while useful, may obscure potential variability in the past. It cannot be assumed the same systems were in play during the mid-Holocene, or indeed characterise the entire region based on a few examples. Again some discussion of this complexity and acknowledgement of the inherent over-simplification of the model is necessary.

- This is a fair criticism - we oversimplified this transition to provide a easily understandable explanation for the apparent delay. Clearly a greater diversify of practices is possible and we should have explicitly acknowledged that. Throughout the revised manuscript, we have replaced "pastoralist" with "pastoralism" when referring to the early populations. We have also altered the title and figure 4b to reflect these changes.

3. *The third issue relates to the population modelling from the existing radiocarbon chronology, based on data published by Manning and Timpson (2014). I appreciate the intent of this communication is not to describe the population model, but it is critical to addressing the issue of causation and the ability to 'refute the hypothesis' tested here.*

- The hypothesis proposed by Wright (2017) is built in part on the suggestion of population increase around 6ka, for which the Manning and Timpson (2014) record is used. We therefore feel it is necessary to demonstrate that the timing of this particular population record does not support the hypothesis. Nonetheless, we provide additional details about this dataset in both the main text (L26-8) and the methods (L334-347).

Human demography is challenging; it can be tempting to use radiocarbon determinations as a proxy for population, both occupation intensity and as a gross estimate of numbers. However, many studies (e.g. Davies et al. 2015) point out issues in the samples to begin with. To my knowledge there has been no systematic critical assessment of the radiocarbon corpus from North Africa, with the exception of a graduate thesis (Phillips 2013). This preliminary work reveals inherent biases in the dataset, concentrations of radiocarbon dates are largely a function of where archaeologists have looked, the number of years a project runs, and presumably budget. This is even before we consider other issues with the dataset, such as those raised by Vermeersch (2014). This is obviously critical to the assumptions made of a population expansion associated with pastoralism. One article is referenced to explain the population increase, but this needs to be dealt with – how are population fluctuations, so critical to the argument, actually empirically demonstrated? A 'dots on map' approach assuming a relationship between population size and the amount of radiocarbon 'dates' is not sufficient in light of recent critiques. The methods section describes the use of summed probability plots from radiocarbon determinations from across North Africa. No information is given on the sampling strategy employed to procure this dataset and this is not sufficiently dealt with in Manning and Timpson (2014). At the very least, some discussion of the critical issues of this analysis should be discussed. In addition more radiocarbon determinations have been published for the region since the 2014 work.

- An effort to highlight these issues has been made in the text (L26-8 in revised manuscript). However, we agree with the reviewer that this particular manuscript is not the most appropriate avenue to publish a revision and update of the Manning and Timpson (2014) dataset. Further work to refine the population modeling is planned. Therefore this manuscript has kept discussion of the subject brief and concise.
- One of us (KM) has already secured funding to investigate the questions raised by this comment. *Peopling the Green Sahara* is a Leverhulme-funded research project. Not only will it collate an updated set of radiocarbon dates, but also it will explore alternate proxies such as organic residues in prehistoric pottery. We envisage gaining a more complete and complex picture of how societies and demographics varied in the Green Sahara. This work is underway, and in our opinion should be kept distinct from the modelling work described here.

Population migration from the eastern Sahara is then suggested have given rise to Pharaonic civilisation; again the radiocarbon chronology upon which this is based is problematic, and recent consideration of archaeological evidence does not adequately support this hypothesis (Wengrow et al. 2014).

- We had felt that by using the verb "appears" in this sentence, we had conveyed our skepticism of this particular claim. This was obviously not the case, and we have rephrased it (L149-150 in the revised manuscript).

Many archaeologists based in North Africa have worked very hard to demonstrate the complexity of human behavioural response to, and impact on, the environment. At this point in the 21st century it would seem more productive to advance theoretical frameworks rather than return to old ones.

- We accept that we could have showed more sophistication in our theoretical framework. We had based our arguments on the framework of Wright (2017) - to demonstrate that even in this framework, his suggestions do not hold. We have revised and expanded the text in (new) section 2 in response to this comment.

3 Reviewer #2

In this study the authors present a modeling study to investigate whether the development of pastoralist societies in Northern Africa during the early and mid Holocene may have had any role in the desertification of the Sahara region at the end of the African Humid period around 5500 years ago. The main conclusion of the work is that the pastoralism may have slowed down the deterioration caused by orbitally-driven climate change rather than enhanced as suggested in Wright et al. (2017).

This is an important message that has broad implications and I totally agree with the authors when stating that Wright et al. provide “an illustrative example of the outdated doctrine against pastoralists”. I actually wonder why Wright et al study received so much attention to even be reported by BBC mundo (<http://www.bbc.com/mundo/noticias-39307995>). It is indeed well known that light-to-moderate grazing actually favors plant species biodiversity, while heavy grazing livestock production can be detrimental and no grazing may also cause it to plummet (Hart, 2001).

- We already had preliminary results from this model when the Wright paper was published. We were rather perplexed about the paper’s conclusion and its coverage too.

My major concern with the present study regards the fact whether the simplified model used by the authors can be used to draw such conclusion, i.e. that the Holocene green-Sahara period may have actually ended earlier without the pastoralism. As the authors also state, more complex climate models still struggle to replicate the Sahara greening during the African Humid Periods, nevertheless I am not fully confident that such a simplified model can be used for the specific question the authors are trying to answer. For example while the model does a good job in replicating the wet and dry periods it is not always in agreement with the sapropel start and end.

- We accept there are some obvious uncertainties in this model and approach. However, we feel it is the best available method to test the bold assertions of Wright (2017). We hope that publishing this study will spur the necessary research across multiple research fields necessary to develop the improved methods requested.

3.1 Minor comments

1) *When submit a manuscript for review is always useful to have the line numbers.*

- Unfortunately, Nature’s L^AT_EX template does not include line numbers. We have rewritten the template to include them now.

First paragraph:

2) *Please change the verb forecast with either simulate or predict*

Main text:

3) *Together with reference 11 and 12 and I suggest to add Hart, 2001.*

4) *Please clarify this sentence: “The model shows several times during the Holocene when northern Africa would have been particularly sensitive to interference (Fig. 4).*

5) *Please clarify and fix this sentence: “Mobility, a distinguishing feature of traditional pastoral systems, promotes periodicity and in the intensity of grazing, which plays a major role in grassland health as grasslands can suffer from undergrazing as much as overgrazing¹¹”. Add also Hart 2001.*

References:

6) *Fix reference 8*

7) *Is reference 36 still a Discussion paper? Please update it.*

- We have made all these corrections.

Methods: 8) Regarding: “The ‘natural’ length of the present interglacial has been estimated from observations³⁶ and potential anthropogenic perturbations highlighted within it²⁸.” The authors do refer to a study, but it would be helpful to have a short summary of how the natural length of the present interglacial has been estimated. The following sentence is also not clear. Why ref. 28 could estimate the natural length of the present interglacial and the authors cannot?

- Fundamentally this comes down to the accuracy of the chronology in relation to the length of the events. We have greatly enhanced the discussion of this analysis in the revised manuscript, which has involved moving some discussion from the methods section. L47-73 are the relevant lines in the revised manuscript.

9) *Change “model of Lui⁸” to read “model of Liu⁸”*

10) *Fix: “The time-varying nature of the forcings mean”.*

11) *Regarding: “Given that sapropel S1 is observed to terminate earlier than the compilation in Fig. 4 A”. It would be good to specify here how much earlier.*

- These 3 corrections and clarifications have been made.

12) Regarding: “It is worth noting that models which are more complicated are not able to adequately simulate the greening of the Sahara”. I would suggest to spend few extra lines in presenting why the CMIP/PMIP models most likely fail in representing the Green Sahara (most likely lack of vegetation and dust feedbacks). I would avoid the use of “It is worth noting” and similar expressions, since if it were not worth noting something, the authors would not write about it.

- We have added two sentences (L90-93) discussing this further in the new section on Robustness.

13) The last two sections “Central Saharan pastoral persistence” and “Misunderstandings propagated by Wright” should be included in the main text.

- We have now elevated these sections into the main text, along with many of the other sections. The Methods section now only consists of a description of the model and the data used to achieve the numerical results.

Acknowledgments: 14) Please correct the name of Zhengyu Liu.

- That is embarrassing. Thank you for spotting that we had spelled Zhengyu’s name incorrectly.

Figures: 15) Figure 1: What is shown in y-axis in both panels? Missing the labels

- We had previously felt that labelling the axes would not have increased the comprehension. This is obviously not a correct assumption. The revised manuscript has the upper panel’s y-axis labelled as “Proxy reconstruction” and the lower panel as “relative population”.

16) Figure 4 missing numbers on the y-axes of panel A and B.

- This oversight has been corrected.

4 Reviewer #3

The authors tackle a very interesting topic: the interaction of pastoralism and ecosystem dynamics in semi-arid regions. They propose that pastoralism could be beneficial for marginal, arid environments, and they list a number of plausible mechanisms to support their proposition. This sounds convincing. However, when put into context with the changes in Holocene Saharan vegetation, the line of arguments appears to be less convincing.

- The model that we have developed finds that the vegetation reduction was delayed. We then try to provide some plausible mechanisms to explain that display. We apologise if our discussion did not convey the uncertainties and limitations of the model sufficiently. In the revised text, we hope that these have become more clear, alongwith the acknowledgment that this model is currently the only tool to test the question.

The title of the paper sounds pretty catchy, addressing the ‘collapse of the green Sahara’. The rapid increase in dust flux from the Sahara into the North Atlantic shelf as seen in the reconstructions by deMenocal et al (2000) and McGee (2013) seems to support the earlier prediction of a tipping point, or collapse, by Brovkin et al. (1998) and Claussen et al. (1999). If one looks at the dust records, however, the increase in dust flux is perhaps an order of magnitude faster than the change in orbital forcing. Hence, geologically speaking, this might be considered a collapse, but a collapse that took many centuries or several dozen generations of humans that once lived in the Holocene Sahara. They may have noticed a very slow increase in aridity, perhaps mainly from hearsay by their ancestors. Figure 2B of this paper supports the very slow changes in the population of the green Sahara specifically around 5,500 years before present, when the rapid change in the Holocene Sahara occurred. Hence, the term “collapse of the green Sahara” in relation to human dynamics in the Holocene Sahara is surely a bit overstated.

- We had chosen the adopt the term that has been most often used to discuss the topic amongst a broad audience, given the readership of the journal. We have changed the title to **Pastoralism may have delayed the end of the green Sahara**, as ‘end’ is ambiguous as to the rate.

The authors support their proposition by showing that their model predicts a somewhat earlier possible termination of the African Humid period than reconstructed. The prediction is internally consistent, the choice of model parameters and the estimate of the robustness of model simulations are also convincing. However, the conclusion that human activity must be the cause for the difference between predicted end of the AHP and number of climate proxies indicating the end the AHP is not compelling. This difference could be attributed to a number of other factors.

- We recognise that this could be for another explanation - with the applicability of the model being the most obvious candidate. We had attempted to address and discuss this in original manuscript, thought obviously not successfully. In the revised manuscript, we have introduced a subsection called "robustness". We had started from the position that it already been proposed that Humans had influenced the system, so it was be logical to address this possibility. In particular, L103-107 have been added directly in response to this comment.

The model used by the authors is fascinating, and it is a great tool for conceptual visualisation. But does it capture the complex dynamics and spatial heterogeneity of the Holocene Sahara? Does it really allow to 'forecast the length of the AHP'? A number of studies, reconstruction (e.g. Shanahan et al., 2015) and modelling (Bathiany et al., 2012, Egerer et al., 2018), have demonstrated that rapid changes in the Holocene Sahara can occur in different regions at different times in the past. Possibly, fast transitions prevailed more in the Western part, and more gradual transitions, in the Eastern part (Brovkin and Claussen, 2008). This complexity is not captured by the conceptual model.

- Clearly a model that considers the region as a single coherent system cannot capture regional variations. We speculate that the different parameter settings may represent different regions. Discussion of the regional aspects were previously only in the methods, but have been promoted to the main text of the revised manuscript (L94-101).

In line with the previous argument: Is the number of proxies indicating the end of the AHP really a proper measure of the end of the AHP? According to the studies mentioned above, the increasing number of indicators for the end of the AHP does not indicate an increased probability of the end of the AHP as a whole, but rather reflects the time-transgressive shift of the AHP termination. Aggregation of geographical heterogeneity is perhaps the weakest argument of this study.

- We accept this criticism, but recognise that at present this is the only option to determine a probabilistic end of the AHP. We have mentioned this facet explicitly in the revised manuscript (L331-333).

I would like to conclude by saying that I like the study as it critically re-assesses and disputes the conventional idea of degradation due to overgrazing. The proposition outlined by the authors might even be valid for the Holocene Sahara. But it is only a hypothesis, not a compelling proof or prediction. Therefore I suggest that this study be published only after careful re-writing and stressing the hypothetical character of the modelling. For example, phrasing the title as a question rather than a seemingly affirmative statement would be one item. Consistently, the hypothetical character of the modelling, specifically its limitation regarding geographical complexity, should be clearly stressed throughout the text.

- We wholeheartedly agree that this is a novel modelling result that should serve as a hypothesis for further research. We have enhanced discussion of the uncertainty in the manuscript - mainly by moving the caveats out of the Methods and into the main text.
- Nature style guides state that titles can contain no punctuation. We feel it is inappropriate to include a question mark - and hence do want to have a question. We have, however, altered the title to **Pastoralism may have delayed the end of the green Sahara** in light of this and other comments.

4.1 Minor comments:

2nd paragraph: The rate of change of precipitation (drying) and in vegetation (desertification) do not necessarily go hand in hand. Liu et al. (2006) differentiate between "stable" and "unstable" collapse which emerge from a loss of stability of just from a fast feedback, respectively. However, this only applies in a statistical sense (Claussen et al., 2013). A single realisation, or record such as the Lake Yoa record, which reveals a gradual decline in rainfall, does not necessarily refute the existence of a bifurcation.

- This is a fair comment. The particular sentence referred has been replaced in the revised manuscript with a longer paragraph that delves into more detail.

2nd paragraph: The term "green Sahara" was surely not coined by Lenton et al. (2008). The Hungarian adventurer László Almásy who found the cave of swimmers the Gilf Kebir wrote about the green Sahara in his book 'The unknown Sahara' in the 1930s. This term was used later in a number of palaeo climate papers in the 1990s.

- We apologise for putting the citation in wrong place in the sentence. We had intended the Lenton et al citation to demonstrate that the green Sahara is a common example of a tipping point. We have removed this citation now.

3rd paragraph: The authors stress that their model – in contrast to the model by Liu et al. (2006/7) – includes the effect of CO2 on rainfall. This effect is, of course, a rather indirect one. CO2 fertilisation might also play a role. It seems that changes in CO2 are important for the tuning of the conceptual model. But for the green Sahara problem it might be rather unimportant.

- We have not been clear in our description of the model. CO2 is really standing in as a metric of the combined impacts of glacial-interglacial cycles - so there is a temperature and precipitation response. This is mentioned on L58.

6th paragraph (or where appropriate): What about fire ecology? People have used fire, and presumably, herders have used fire in a different way and intensity than early farmers did.

- Whilst this is an interesting point, there is little suggestion of early farming in the region. Nonetheless, a change in fire ecology could have potentially resulted as the proportion of pastoralism increased. We suspect that this was a minor contribution, and do not discuss it in the revised manuscript.

Penultimate paragraph: I do not understand the last sentence. Which teleconnection is meant? Perhaps, the link between Holocene pastoralists in the Sahara and the global carbon cycle is a bit far-fetched.

- We agree that it is certainly speculative. However Ruddiman (2013) proposes an anthropogenic variation in the carbon cycle at precisely this time. We therefore feel that we should discuss how our research may relate to this - variations in Congo peat storage seems the only possibility. We have revised the sentence to convey the speculative nature of this (L176-177).

At a number of places, the authors emphasise that global, comprehensive models do not properly describe the Holocene greening of the Sahara. Indeed, most comprehensive models yield a small greening (although the first 'green model' (Claussen and Gayler, 1997) shows a substantial greening). However, this statement does not imply that the conceptual model of the 'Brovkin/Liu type' is superior. Both types of models serve different purposes.

- We had not meant to imply that our idealised model was superior. Rather we wanted to stress that it would not be better to attempt a forecast of the natural end of the Holocene African Humid Period with fully coupled GCMs. We have added some sentences to explain this (L90-93).

REVIEWERS' COMMENTS:

Reviewer #1 (Remarks to the Author):

The authors have very thoroughly and thoughtfully considered the comments provided. While there are still issues to be resolved, I appreciate this is not possible given the limitations of this short communication amongst other inherent problems with the approach that cannot be addressed here. In sum, as the authors state, the purpose of this paper is to stimulate further discussion, which is undoubtedly will, and so should be published.

Rebecca Phillipps

Reviewer #2 (Remarks to the Author):

The authors have addressed my comments and did a better job in framing the study. I therefore recommend it for publication after few additional minor comments have been taken into account.

Minor Comments:

Figures:

Fig. 2: I am still not satisfied with this figure.

The authors added some very vague y axis and the caption is still unclear.

"Reconstructions of Holocene northern Africa": Reconstructions of what?

What is the order of proxy reconstructions in panel A? Are they ordered according to latitude? If there is no specific reason I would suggest to group them by colors? Eg. Orange (top), Red, Blue (and then below the others) to sort of follow panel B.

The y of panel A could be Paleoclimate Reconstructions (wet/dry periods)

Panel B. Relative population is in %? Goes from 0 to 1 where 1 would be the max?

Fig. 3: The red horizontal line is difficult to see (the color), I suggest to use a dashed line.

"The use of a geochemical index to identify sapropels (such as the Ba/Al ratio used in Fig 2B) minimises the impact of post-depositional oxidation⁵⁸ that has been shown to remove the uppermost part of a sapropel."

This sentence has nothing to do with the caption that should be kept concise and just described what is shown.

Also sentences like "This chronology suggests that the most recent sapropel was of much shorter duration than previous instances, yet only includes one other interglacial sapropel (at 129.5 ka)."

or "This highlights potential issues with the chronology around 400 and 800 ka, rather than suggesting abnormalities in the Holocene instance. These two interglacials have orbital configurations most like

the Holocene³³, but appear to show sapropel onsets 90_ out of phase with all the other occasions.", belongs to the main text and should be moved there unless the authors deem that are absolutely necessary in the caption to explain what is shown.

Fig. 5: What is the grey dashed line?

Again no scale in the y-axis: it's relative but relative to what? I assume there should be a 1 and 0 somewhere.

In Panel C there are numbers but not units! 150 what? It needs to be specified. Same in figures S1b and S2.

Text:

L28 African Humid Period has already been mentioned above (L19), therefore the authors should

define the acronym (AHP) and should make sure to use capital letters.

L41 The authors should add also the Tierney et al., 2017 (Rainfall regimes of the Green Sahara) as citation together with reference 25.

LL50-51: "Mediterranean sapropel deposition is used as an indicator of humid conditions in northern Africa, because they have some of the best chronologies and so highlight the limits of the approach." Please clarify this sentence.

L93: In ref. 37, Pausata et al. 2017 show that the poor capturing of vegetation and dust feedbacks may also be able to explain the lack of proxy-model agreement in the ENSO region. Regarding exclusively the Sahara region the authors should refer to Pausata et al. 2016 (Impacts of dust reduction on the northward expansion of the African monsoon during the Green Sahara period) and include "dust feedbacks" when mentioning the cause of long standing biases.

L103 use the acronym AHP.

L131-132 I suggest to rewrite the sentence "The historical ..." to better connect the sentences. Something along those lines:

"Wright uses historical analogues such as the Rapa Nui environmental degradation and ... that appear inappropriate. For example,"

LL134-135 I am not sure I understand what the authors meant to say. As it is written it almost seems that the European contact that eventually caused the collapse of Rapa Nui people. The decline of the Rapa nui society started well before European discovery in 1722 and it was caused by the over-exploitation of the island's environment, most notably through deforestation of almost all the island's trees.

From citation 43: "The most continuous record of the last millennia obtained so far is from Lake Raraku. This record encompasses the last ~3700 cal y BP, with a minor sedimentary gap of less than 700 years at the interval of interest (Cañellas-Boltà et al., 2013) (Figure 1). This gap represents a reduction of 80–90% in the missing interval with respect to previous works (Flenley et al., 1991; Mann et al., 2008; Sáez et al., 2009). As a result, it has become clear that the replacement of palm forests by grasslands took place in a gradual manner between approximately 500 BC and AD 1500 (Figure 2). During these 2000 years, the forest decline and the progressive establishment of grasslands proceeded at a fairly constant rate (ca. 3% reduction per century, in palm pollen units), with some minor accelerations. The onset of this replacement roughly coincided with the first appearance of *V. litoralis* (Figure 2), which has been used as indirect evidence for island colonization. Hence, it could be hypothesized that human activities played a role in the palm forest demise since initial colonization but in a gradual rather than catastrophic manner. This hypothesis is also supported by Mulrooney's (2013) evidence of continuous settlement and gradual landscape use."

So I am not sure I understand what the Europeans had to do with the degradation of the island. In any case, I agree that the Rapa Nui case doesn't have much to do with the Sahara case, but I suggest to clarify and better contextualize these sentences.

Finally carefully check the references: e.g. ref 10; ref. 24 (capital letters and "); ref. 25 (capital letters); ref. 37 (capital letters) etc.

Reviewer #3 (Remarks to the Author):

The authors replied comprehensively to my comments and revised their manuscript accordingly. Therefore I recommend the publication of the manuscript.

FIRST PARAGRAPH

Reviewer 1 and 3 did not raise further comments that needed addressing. Reviewer 2 made some further points. They start by saying “The authors have addressed my comments and did a better a job in framing the study. I therefore recommend it for publication after few additional minor comments have been taken into account.” We thank the reviewer for this opinion and their explicit statement. We also appreciate effort and diligence devoted to catching our mistakes and any sloppiness.

Fig. 2

The reviewer states they are not satisfied with this figure. They state the following reasons.

1. The authors added some very vague y axis and the caption is still unclear: ”Reconstructions of Holocene northern Africa”: Reconstructions of what? The y of panel A could be Paleoclimate Reconstructions (wet/dry periods)
2. What is the order of proxy reconstructions in panel A? Are they ordered according to latitude? If there is no specific reason I would suggest to group them by colors? Eg. Orange (top), Red, Blue (and then below the others) to sort of follow panel B.
3. Panel B. Relative population is in %? Goes from 0 to 1 where 1 would be the max?

This figure has been edited. We have altered both y-axes titles. We have also added some sums at the bottom of panel A, to better explain how we arrived at the time series in Fig. 5 from this plot. (1) We have adopted the Reviewer’s suggestion for the y-axis title of panel B. (2) They are presented in order of termination date. This is now stated in the caption (along with an explanation of the new summation values. (3) We have changed the y-axes to ”summed probability density, as in the original publication of Manning & Timpson (2014). As this is a probability density, its integral equals 1. We feel that presenting the actual values would cause more confusion than it clears up, especially as the respective axes would overlap.

Fig. 3

1. Fig. 3: The red horizontal line is difficult to see (the color), I suggest to use a dashed line.
2. “The use of a geochemical index to identify sapropels (such as the Ba/Al ratio used in Fig 2B) minimises the impact of post-depositional oxidation⁵⁸ that has been shown to remove the upper-most part of a sapropel.” This sentence has nothing to do with the caption that should be kept concise and just described what is shown.
3. Also sentences like “This chronology suggests that the most recent sapropel was of much shorter duration than previous instances, yet only includes one other interglacial sapropel (at 129.5 ka).” or “This highlights potential issues with the chronology around 400 and 800 ka, rather than suggesting abnormalities in the Holocene instance. These two interglacials have orbital configurations most like the Holocene³³, but appear to show sapropel onsets 90 out of phase with all the other occasions.”, belongs to the main text and should be moved there unless the authors deem that are absolutely necessary in the caption to explain what is shown.

This overly-detailed caption was a left-over from when the figure was intended to acts as a stand-alone diagram in the Supplementary Information. The caption has been shortened, by moving important points into the main text, and removing points that were not strictly necessary. We have additionally increased the visibility of the red line as suggested. Incidentally, the colored lines in this figure using a color-blindness tool to check that the shade are still differentiable.

Fig. 5

1. Fig. 5: What is the grey dashed line?

This reference has now been added

- LL50-51: “Mediterranean sapropel deposition is used as an indicator of humid conditions in northern Africa, because they have some of the best chronologies and so highlight the limits of the approach.” Please clarify this sentence.

We have removed the clause about the limits of the approach. We replaced “best” with most accurate to clarify our assessment criteria.

- L93: In ref. 37, Pausata et al. 2017 show that the poor capturing of vegetation and dust feedbacks may also be able to explain the lack of proxy-model agreement in the ENSO region. Regarding exclusively the Sahara region the authors should refer to Pausata et al. 2016 (Impacts of dust reduction on the northward expansion of the African monsoon during the Green Sahara period) and include “dust feedbacks” when mentioning the cause of long standing biases.

This has been corrected.

- L103 use the acronym AHP.

Thanks. That reads better.

- L131-132 I suggest to rewrite the sentence “The historical . . .” to better connect the sentences. Something along those lines: “Wright uses historical analogues such as the Rapa Nui environmental degradation and ... that appear inappropriate. For example,”

This sentence has been edited after this suggestion

- LL134-135 I am not sure I understand what the authors meant to say. As it is written it almost seems that the European contact that eventually caused the collapse of Rapa Nui people. The decline of the Rapa nui society started well before European discovery in 1722 and it was caused by the over-exploitation of the island’s environment, most notably through deforestation of almost all the island’s trees. From citation 43: *The most continuous record of the last millennia obtained so far is from Lake Raraku. This record encompasses the last 3700 cal y BP, with a minor sedimentary gap of less than 700 years at the interval of interest (Cañellas-Boltà et al., 2013) (Figure 1). This gap represents a reduction of 80–90% in the missing interval with respect to previous works (Flenley et al., 1991; Mann et al., 2008; Sáez et al., 2009). As a result, it has become clear that the replacement of palm forests by grasslands took place in a gradual manner between approximately 500 BC and AD 1500 (Figure 2). During these 2000 years, the forest decline and the progressive establishment of grasslands proceeded at a fairly constant rate (ca. 3% reduction per century, in palm pollen units), with some minor accelerations. The onset of this replacement roughly coincided with the first appearance of *V. litoralis* (Figure 2), which has been used as indirect evidence for island colonization. Hence, it could be hypothesized that human activities played a role in the palm forest demise since initial colonization but in a gradual rather than catastrophic manner. This hypothesis is also supported by Mulrooney’s (2013) evidence of continuous settlement and gradual landscape use.* So I am not sure I understand what the Europeans had to do with the degradation of the island. In any case, I agree that the Rapa Nui case doesn’t have much to do with the Sahara case, but I suggest to clarify and better contextualize these sentences.

There is an ongoing discussion amongst the community about the transitions on Rapa nui. The Reviewer is right to highlight this discussion. Although we believe our statement to be correct, we have removed it completely as it doesn’t really matter for our point.

- Finally carefully check the references: e.g. ref 10; ref. 24 (capital letters and “); ref. 25 (capital letters); ref. 37 (capital letters) etc.

The references have now been checked and corrected. Thank you for spotting these issues.